

# What's anomalous in LHC jets?

Thorsten Buss[1], Barry M. Dillon[1], Thorben Finke[2], Michael Krämer[2],
Alessandro Morandini[2], Alexander Mück[2], Ivan Oleksiyuk[2] and Tilman Plehn[1]

**1** Institut für Theoretische Physik, Universität Heidelberg, Germany
**2** Institute for Theoretical Particle Physics and Cosmology (TTK),
RWTH Aachen University, Germany

## Abstract

Searches for anomalies are a significant motivation for the LHC and help define key analysis steps, including triggers. We discuss specific examples how LHC anomalies can be defined through probability density estimates, evaluated in a physics space or in an appropriate neural network latent space, and discuss the model-dependence in choosing an appropriate data parameterisation. We illustrate this for classical k-means clustering, a Dirichlet variational autoencoder, and invertible neural networks. For two especially challenging scenarios of jets from a dark sector we evaluate the strengths and limitations of each method.

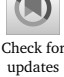
doi:10.21468/SciPostPhys.15.4.168

# 1 Introduction

Searches for new physics at the LHC are traditionally defined by testing theory hypotheses and comparing them to Standard Model predictions using likelihood methods. Weaknesses of this approach are that search results are hard to generalize, and that we still live in fear of having missed a discovery by not asking the right questions. Modern machine learning (ML) offers a conceptual way out of this dilemma through anomaly searches. Looking at LHC physics these concepts can be developed most easily for QCD jets, analysis objects available in huge numbers with much less physics complexity than full events. Nevertheless, they are complex enough such that possible new physics signatures can hide in a non-trivial way.

Considering the task of finding anomalies in these jets, unsupervised ML methods are favored. In contrast to supervised methods, unsupervised methods do not rely on labeled data. This allows for sensitivity to a broad range of potential signal models and the application directly on data. Autoencoders(AEs) are a simple unsupervised ML-tool relying on a bottleneck in a mapping of a data representation onto itself, and constructing a typical object. They have been shown to identify anomalous jets in a QCD jet sample, indicating that anomaly searches at the LHC are a promising new analysis direction for the upcoming LHC runs [1, 2]. Autoencoders do not induce a structure in latent space, so we have to rely on the reconstruction error as the anomaly score. This corresponds to a definition of anomalies as an unspecific kind of outliers. The conceptual weakness of autoencoders becomes obvious when we invert the searches, for instance searching for anomalous QCD jets in a top-jet sample [3]. This failure in symmetric searches leads us to the more fundamental question how we define anomalous jets or events at the LHC and what kind of anomaly measure captures them [4].

Moving beyond purely reconstruction-based autoencoders, variational autoencoders [5] (VAEs) add structure to the latent bottleneck space. In the encoding step, a high-dimensional data representation is typically mapped to a low-dimensional latent distribution, from which the decoder learns to generate new high-dimensional objects. The latent bottleneck space then contains structured information which might not be apparent in the high-dimensional input representation. Again, VAEs have been shown to work for anomaly searches with LHC jets [6, 7], and we can replace the reconstruction loss by an anomaly score defined in latent space. At this point, an interesting and relevant question becomes the choice of latent space and specific anomaly score, for instance in a Dirichlet latent space [8, 9] which encourages mode separation [4].

ML-methods for anomaly detection at the LHC have generally received a lot of attention in the context of anomalous jets [10–17], anomalous events pointing to physics beyond the Standard Model [18–35], or enhancing established search strategies [36–42]. They include a first ATLAS analysis [43], experimental validation of some of the methods [44, 45], quantum machine learning [46], applications to heavy-ion collisions [47], the DarkMachines challenge [48], and the LHC Olympics 2020 community challenge [49, 50].

In spite of this wealth of possible practical applications, the fundamental question still needs to be studied, namely *what defines an anomaly search at the LHC?* For large and stochas-

tic datasets, the concept of outliers is difficult to define unambiguously, because any LHC jet or event configuration will occur with a finite probability, especially after we include detector imperfections [51–55]. In this situation, a simple, working definition of anomalous data is an event which lies in a low-density phase space region. Such a phase space region can be defined based on any set of kinematic observables, notably including a latent space variable constructed by a neural network. While such a general definition cannot be understood directly using quantum field theory, it can be linked to first-principles through the corresponding simulations. An alternative definition based on overdensities can be applied for localised signals. These signals are typically localised in some global observable like the invariant mass of the events. The background density then needs to be inferred e.g. through sideband methods. Weakly supervised methods such as the classification without labels (CWoLa) method [56] can be applied in these settings to learn a likelihood ratio classifier for an anomalous signal in various forms [38,42,51].

For anomaly searches at the LHC we probe and encode the phase space probability of the known background through a set of jets or events. This training of the anomaly-search network can use simulations of a pure background dataset, or it can use actual data under the assumption that a small signal contamination will be ignored by the respective network training and architecture. Once the background density is encoded in the network, the goal of the anomaly detection methods discussed in this work is to derive an anomaly score for each data point, based on the learned background density. This way, our anomaly detection methods are fundamentally linked to density estimation, which in turn depends strongly how we define the observables in the analysis. They are also closely linked to standard LHC search strategies, where we test and rule out background hypotheses without reference to a specific signal hypothesis. Challenges to this density-based anomaly detection using machine learning are the high dimensionality of the feature vectors describing LHC events, or the fact that the probability densities in the latent and phase spaces are not invariant under reparameterizations or reweighting of the input data and the choice of network architectures. In this paper we will study three different ways of defining an anomaly score for LHC jets based on density estimation, to illustrate the challenges and advantages of different approaches and network architectures.

As reference datasets we introduce two dark-matter-inspired jet signatures in Sec. 2. The underlying new physics model is Hidden Valleys, made of a strongly interacting, light dark sector [57–59]. Our physics task is similar to Ref. [17], where the technical focus is on a fixed signal type and set of observables. When we produce particles from this dark sector, they can decay within the dark sector and form a dark shower, but this dark shower will eventually switch to SM fragmentation and turn into a semi-visible jet [60–65] or a pure, modified QCD jet [1]. In both cases we can use ML-based subjet tools to tag the signal jets, provided we know and control the signal hypothesis. The problem is that the model parameter space is too large to rely on standard hypothesis testing; there are also reasons to doubt that the dark sector showering modelled in Pythia [66] is accurate due to differences between the strong sector in the SM and in the dark sector. So our strategy will be to search for such dark jets using anomaly detection on jets. We note that both our signals are particularly hard to distinguish from QCD jets, unlike fat jets arising from Standard Model decays.

Facing the task of extracting our two dark jets signals from QCD jets in an unsupervised, data-driven setup we will discuss and benchmark three different methods. Our first approach is based on classic *k-means clustering* combined with density estimation, and does not require modern deep learning. In Sec. 3 we introduce a general setup suitable for stochastic datasets and show how the anomaly score can be optimized for generic classes of anomalies. When it comes to the best-performing latent spaces, we found that *Dirichlet VAEs* (DVAEs) outperform for example standard VAEs or latent Gaussian mixture models for hadronically decaying,

boosted top jets as anomaly [4]. In Sec. 4 we show how our two signal jet samples benefit from a better choice in preprocessing, and how the jet representations as jet images and energy flow polynomials (EFPs) [67] compare in terms of anomaly searches. Our third method uses *normalizing flow networks* [68, 69], specifically invertible neural networks (INNs) [70, 71], to bijectively map phase space to latent space. To limit the dimensionality of this mapping we use EFPs as the phase space representation. INNs are the cleanest way to directly estimate the density of the jets in phase space (or, physics space). They learn an invertible transformation from the phase space of a jet to a multi-dimensional Gaussian along with a Jacobian that ensures the density is properly accounted for in the transformation. This gives us both a structured Gaussian latent space for the jets, and a method to estimate the density of the jets in the physics space. We want to study the use of densities as anomaly scores. Comparing these three fundamentally different concepts on the two dark jets samples we find very similar performance and a sizeable dependence on the preprocessing of the respective datasets, specifically the reweighting of the inputs.

## 2 Dark jets

Fat jets from boosted, hadronically decaying particles are among the most complex objects entering LHC analyses. Produced from a single relativistic particle and undergoing decay, showering, and hadronization, they contain between 20 and 100 relevant constituents per jet, in which we need to identify the crucial decay and showering patterns. In the Standard Model, established fat jet signatures arise from boosted top quarks, boosted weak bosons, and boosted Higgs bosons. In addition, many interesting new physics scenarios can lead to fat jets. One class of such models are Hidden Valley models [57–59].

The variety of Hidden Valley models and their parameters is extensive, for our purpose their leading feature is a strongly coupled $SU(3)_d$ dark sector with fermions coupling to the SM through some portal. Jets in these models can be produced from new physics states with couplings to the SM and dark sectors, so the showering involves radiation into the dark sector. The resulting jets are referred to as semi-visible jets. However, not all jets from Hidden Valley models result in a significant showering of stable dark hadrons, so we refer to them more generally as dark jets. Because LHC experiments cannot search for the full range of possible anomalous jets using theory-based hypothesis methods, such searches are a clear candidate for an unsupervised machine-learning treatment, similar to strategies proposed for SUEP signals [72]. Before we proceed with the different anomaly-search strategies, we briefly describe the two signal datasets we use for our analysis, including possible data representations and data preprocessing.

### 2.1 Datasets

To cover the range of challenges and solutions in density-based anomaly searches, we define two signal benchmarks with different underlying physics features:

1. Aachen dataset

$$pp \rightarrow Z' \rightarrow \bar{q}_d q_d \,, \quad \text{with} \quad m_{Z'} = 2 \,\text{TeV}, \quad \text{and} \quad m_{q_d} = 500 \,\text{MeV}\,. \quad (1)$$

   The $Z'$ mediator to the dark sector is described by a weakly interacting $U(1)'$ gauge group, and the dark quarks are charged under a strongly coupled dark sector and this $U(1)'$. Like QCD, the dark sector contains dark pions, $m_{\pi_d} = 4$ GeV, and dark rho mesons, $m_{\rho_d} = 5$ GeV. The $Z' - \rho_d^0$ mixing leads to the decay of the $\rho_d^0$ to SM-quarks. The other dark mesons are stable and escape the detector. The fraction of constituents escaping

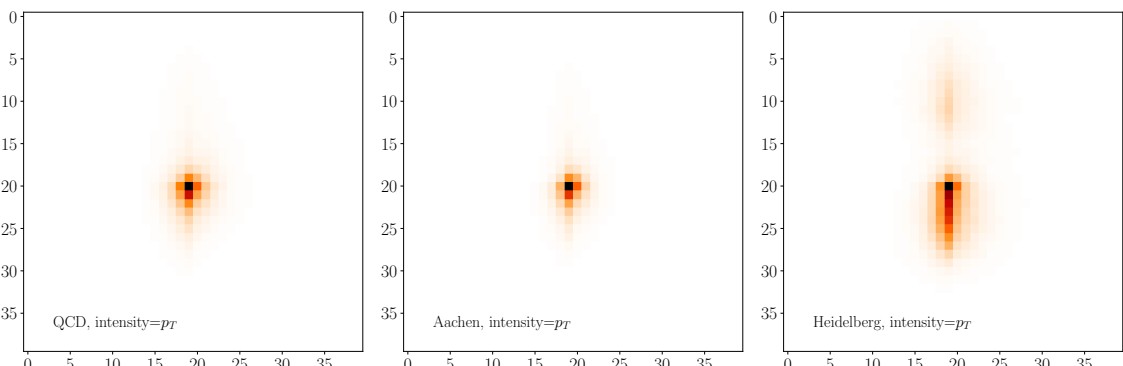

Figure 1: Averaged preprocessed jet images for the QCD background and for the two dark-jet samples.

the detector is $r_{\text{inv}} = 0.75$. This scenario is a typical semi-visible jet, as explored in detail in [3,65].

2. Heidelberg dataset

$$pp \rightarrow \bar{q}_d q_d\,, \qquad \text{with} \qquad m_{q_d} = 50 \text{ GeV}\,. \qquad (2)$$

The dark quarks are charged under $SU(3)_c$ and the dark $SU(3)_d$, so after pair-production the dark quarks will radiate into the SM and dark sectors. Eventually, the dark states decay back to SM particles plus a dark boson $b_v$ with mass 5 GeV. This dark boson hadronizes to scalar and pseudo-scalar dark meson states with masses assumed to be 10 GeV. For our choice of model parameters, the dark mesons decay back to the SM, i.e. $r_{\text{inv}} \simeq 0$ for the Heidelberg dataset [1].

The Aachen dataset is simulated using Madgraph5 [73] for the hard process and the Hidden Valley model [74, 75] in Pythia8.2 [66] for showering and hadronization. The Heidelberg dataset is simulated using just the Hidden Valley model in Pythia8.2. The light QCD background jets are simulated using MadGraph5 to obtain di-jet events and Pythia8.2 for showering and hadronization. For a fast detector simulation we rely on Delphes3 [76]. A background which we do not consider here arises from detector malfunctions such as dead cells, however this is not modelled by Delphes so we are unable to implement it here. Nevertheless this does not alter the core results of the analysis. The jets are reconstructed using the anti-$k_T$ algorithm [77] with $R = 0.8$ in FastJet [78] and fulfill

$$p_{T,j} = 150 \ldots 300 \text{ GeV}\,, \qquad \text{and} \qquad |\eta_j| < 2\,. \qquad (3)$$

Although these parameters and cuts do not guarantee that all decay products of the Heidelberg dark quarks end up in the same jet, in many cases they will. These two signal benchmarks allow us to probe different aspects of dark jets, the Aachen dataset providing a typical example of semi-visible jets and the Heidelberg dataset providing a typical example of decay-like or two-prong-like new physics jets.

To represent the jets we use the standard jet images or calorimeter images, replacing the usual high-level observables by the direct detector output and applying basic preprocessing [79–83]. The size of the images is $40 \times 40$ pixels. The key preprocessing steps are centering and normalizing the pixels jet by jet, to sum to one. In Fig. 1 we illustrate the different patterns using averaged jet images after centering them and rotating the main axis to 12 o'clock. In this representation, the semi-visible nature of the Aachen dataset can hardly be distinguished

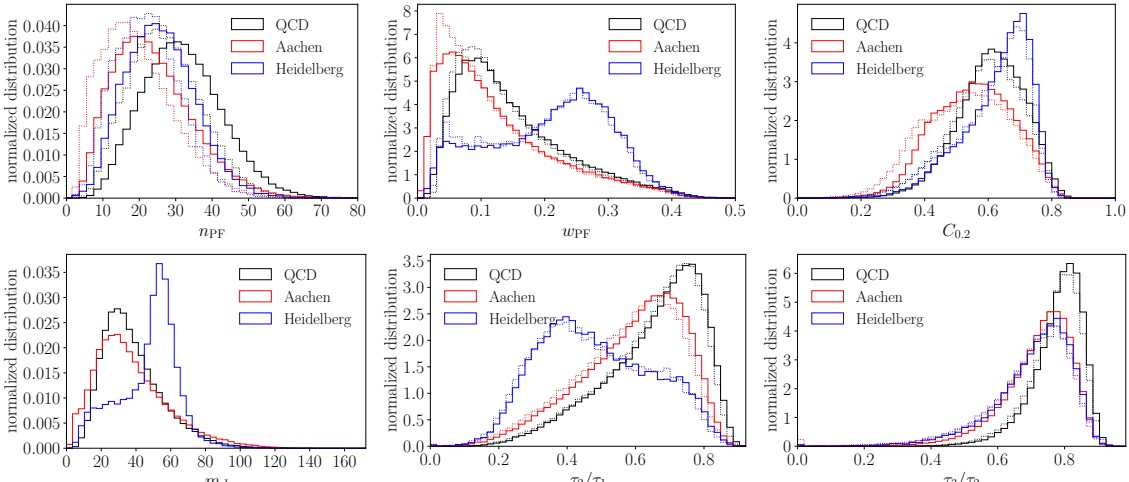

Figure 2: High-level observables calculated from the four-vectors of the constituents (solid) and the calorimeter pixels (dotted) comparing the QCD jets with the Aachen and Heidelberg dark jets.

from the QCD background, while the Heidelberg dataset shows its two-prong structure with a typical distance between the prongs induced by the window in the transverse momentum.

To illustrate the physics behind the QCD and signal jets, we can use standard subjet observables. First, we count the number of jet constituents ($n_{\text{PF}}$) and compute its radiation distribution or girth ($w_{\text{PF}}$) [84], which can be reconstructed from particle-flow output or from jet images. In addition, the two-point energy correlator $C_{0.2}$ is known to separate for instance quark and gluon jets [85],

$$n_{\text{PF}} = \sum_i 1 \,, \qquad w_{\text{PF}} = \frac{\sum_i p_{T,i} R_{i,\text{jet}}}{\sum_i p_{T,i}} \,, \qquad C_{0.2} = \frac{\sum_{ij} p_{T,i} p_{T,j} (R_{ij})^{0.2}}{\left(\sum_i p_{T,i}\right)^2} \,. \qquad (4)$$

We show these observables in Fig. 2, both calculated from the jet constituents and from the pixelized jet image. While the number of soft constituents is similar for all three samples, with slightly smaller numbers for the semi-visible jets, the Heidelberg dark jets show clear signs of massive decays and separated prongs. Obviously, the same pattern is visible in the jet mass, which in this case reflects our choice of $m_{q_d} = 50$ GeV. Because preprocessing of the jet images includes normalizing the pixel entries, we can extract the jet mass only from the constituents. The shoulder towards smaller jet masses corresponds to jets where we lose hard constituents.

Finally, we can track the number of prongs through the IR-safe N-subjettiness ratios [86], which should agree between constituent-based and image-based definitions. Only the Heidelberg dataset shows a significant deviation from the democratic limit, $\tau_i/\tau_j \simeq 1$, in the ratio $\tau_2/\tau_1 \simeq 0.4$, indicating a 2-prong structure in a fraction of its dark jets. The slight difference between the constituent-based and image-based high-level observables is easily explained by the finite resolution of the jet images and the lack of IR-safety for the number of constituents. The jet mass and the N-subjettiness show the smallest effects, as one would expect by construction. We also emphasize that a sensitivity to soft and collinear splittings is only problematic when we compare high-level observables to perturbative QCD predictions, but not when we train supervised or unsupervised classification networks on simulations or data.

## 2.2 Energy flow polynomials

Energy flow polynomials (EFPs) [67] provide a powerful systematic description of the phase space patterns of jets, described by the constituents' transverse momenta and their geometric separation,

$$z_i = \frac{p_{T,i}}{p_{T,J}}, \qquad R_{ij} = \sqrt{(\Delta y_{ij})^2 + (\Delta \phi_{ij})^2}. \tag{5}$$

We use prime EFPs, which are not a product of EFPs with lower rank, and remove the constant EFP. For a maximum of order three in the angular distance $R_{ij}$ this leaves us with the eight EFPs

$$\text{EFP1} = \sum_{i_1,i_2} z_{i_1} z_{i_2} R_{i_1 i_2}, \qquad\qquad \text{EFP5} = \sum_{i_1,i_2,i_3} z_{i_1} z_{i_2} z_{i_3} R_{i_1 i_2} R_{i_1 i_2} R_{i_1 i_3},$$

$$\text{EFP2} = \sum_{i_1,i_2} z_{i_1} z_{i_2} R_{i_1 i_2} R_{i_1 i_2} \approx 2\frac{m_J^2}{p_{T,J}^2}, \qquad \text{EFP6} = \sum_{i_1,i_2,i_3} z_{i_1} z_{i_2} z_{i_3} R_{i_1 i_2} R_{i_2 i_3} R_{i_1 i_3},$$

$$\text{EFP3} = \sum_{i_1,i_2} z_{i_1} z_{i_2} R_{i_1 i_2} R_{i_1 i_2} R_{i_1 i_2}, \qquad \text{EFP7} = \sum_{i_1,i_2,i_3,i_4} z_{i_1} z_{i_2} z_{i_3} z_{i_4} R_{i_1 i_2} R_{i_1 i_3} R_{i_1 i_4},$$

$$\text{EFP4} = \sum_{i_1,i_2,i_3} z_{i_1} z_{i_2} z_{i_3} R_{i_1 i_2} R_{i_1 i_3}, \qquad \text{EFP8} = \sum_{i_1,i_2,i_3,i_4} z_{i_1} z_{i_2} z_{i_3} z_{i_4} R_{i_1 i_2} R_{i_2 i_3} R_{i_3 i_4}. \tag{6}$$

Depending on the application, we can reweight the relative importance of the momentum fraction and the angular separation by replacing

$$z_i \to z_i^\kappa, \qquad \text{and} \qquad R_{ij} \to R_{ij}^\beta, \tag{7}$$

with appropriate values for $\kappa$ and $\beta$. In Fig. 3 we show the first six EFPs for the signal and background datasets. As for the high-level observable, we see a clear difference between the QCD and Aachen datasets on the one hand and the Heidelberg dataset on the other. This difference is linked to the two-prong structure and the finite jet mass of this signal sample. We can understand the signal pattern in the Heidelberg sample for EFP2, where from the threshold for the jet kinematics in Eq.(3) we can estimate

$$\frac{m_J}{p_{T,J}} \lesssim \frac{50 \text{ GeV}}{150 \text{ GeV}} \qquad \Rightarrow \qquad \text{EFP2} \lesssim \frac{2}{9}. \tag{8}$$

The fact that the same pattern appears in almost all EFPs indicates that the EFPs are strongly correlated.

## 2.3 Preprocessing

For our analysis we can apply various preprocessing steps to the jet images, some of them already outlined in Ref. [79], and some of them with the specific goal of enhancing the sensitivity of a given anomaly detection method. In general, we also expect such preprocessings to affect the sensitivity to specific physics signals [17,87].[1]

The first choice, crucial for any neural network, is how we define the $p_T$-information per jet constituent. A naive jet image representation typically uses the sum of the constituent $p_T$ as the pixel value. Both Standard Model and anomalous features in jets occur at very different $p_T$-scales; hard decays lead to features at $p_T \gg 1$ GeV, while jets with a modified parton

---

[1] We note also that the preprocessing could be replaced by learning representations that are invariant to symmetry transformations and augmentations of the data using self-supervised contrastive learning, as in JetCLR [88].

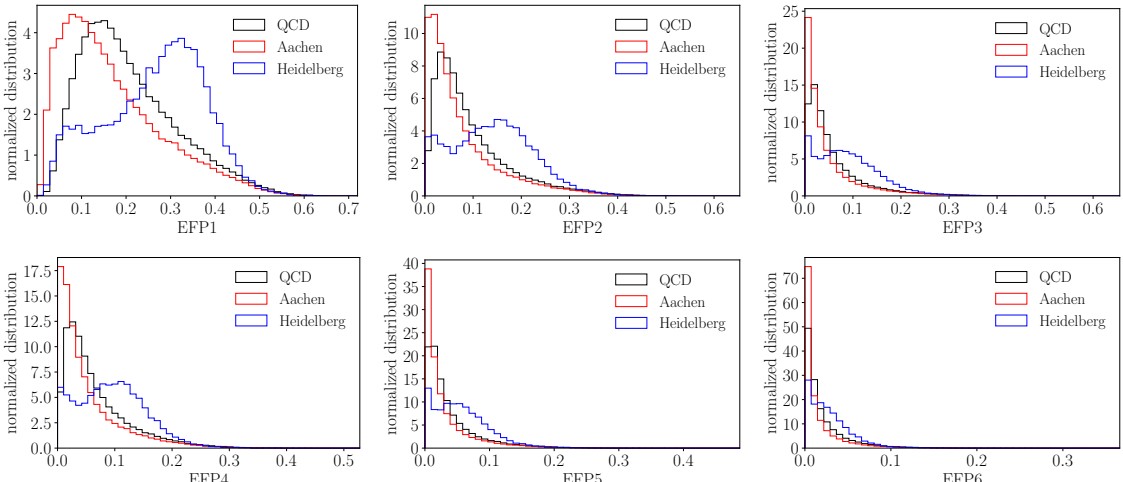

Figure 3: Leading EFPs for QCD jets and for the Aachen and Heidelberg dark jets, computed from the jet constituents.

showering are sensitive to GeV-scale constituents, similar to quark-gluon tagging. This means the standard choice biases a classification or anomaly detection technique towards features at high $p_T$ [3, 87]. This explains why autoencoders tag jets with higher complexity more easily if complexity or structure is usually assumed to affect the harder features of the jet. However, the two signal datasets in this paper fall into two different categories, the Heidelberg sample being more and the Aachen sample being less complex than the QCD background.

For our preprocessing we start by defining a dimensionless input and normalize each image such that the total intensity summed over all pixels is one. For typical loss functions, compressing a wide range of (input) values leads to an improved numerical performance. To study and exploit a potential bias, we consider different choices to the pixel intensity in the jet image,

$$p_T \rightarrow p_T^n, \qquad \text{with} \qquad n \in (0, 1]. \tag{9}$$

Established working points include square root reweighting ($n = 1/2$) or $n = 1/4$ [3]. Aside from the obvious choice of the pixel transverse momentum, the alternative reweightings stretch the resolution at low transverse momenta and move the peak of the pixel distribution to higher reweighted intensity values. This allows the network to extract more information from the large number of soft pixels, while keeping most of the information on hard, decay related pixels. The reweightings also change the density of the jets in physics space, and given that we define anomalous jets as those in the low density regions, they physically alter what jets are anomalous. For an optimal network training it might eventually be beneficial to provide the network with two reweighted inputs, one focusing on soft pixels and one focusing on hard pixels [87].

Second, we need to deal with the sparsity of the jet images. We use a Gaussian filter to decrease the sparsity, convoluting the image with a Gaussian smearing kernel with the width $\sigma = 0.5 \dots 3.0$ pixels. This filter also correlates neighbouring pixels. In Fig. 4 we illustrate the effect of the $p_T$-reweighting and the Gaussian filter on a single QCD jet image. The smearing is equivalent to using a correspondingly defined kernel MSE loss [3] and provides a better measure of similarity for jet images than employing the standard mean squared error (MSE) distance measure. If the intensity in a bright pixel is shifted to a neighbouring pixel, the resulting image is closer to the original image than an image where the intensity is moved further away. Note that neither the reweighting of the pixel intensity nor the application of the Gaussian kernel destroys information. The reweighting is done with a bijective mapping.

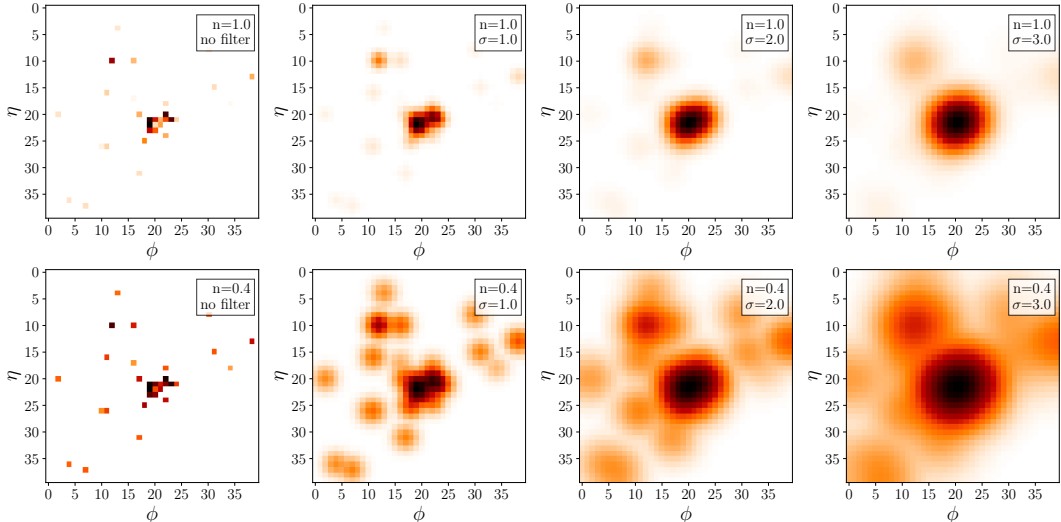

Figure 4: Illustration of two different $p_T$ reweightings (top and bottom) and the Gaussian filter with a varying width $\sigma$, applied to a single QCD jet image.

The Gaussian kernel applied is fixed and contains no randomness. It is therefore invertible except for edge effects where intensity can be smeared out of the image. However, we expect these effects to be negligible, as the image range in $\Delta\eta$ and $\Delta\Phi$ is sufficiently large compared to the jet radius.

For networks working on EFPs we can apply similar preprocessing steps, either reweighting the momentum fractions $z_i$ individually or reweighting the EFPs as a whole. To exploit some of the analytic properties of the INN we choose the latter option, including a reweighting $\text{EFP}_i \to \log \text{EFP}_i$. The Gaussian filter is not relevant for the EFP representation. To estimate the correlations between the different EFPs we turn to Fig. 5. Indeed, the EFPs we use as network input are strongly correlated in a non-linear way. Because of these correlations, for instance the QCD jets and the Heidelberg dark jets have a similar overall structure in the 8-dimensional EFP space, but populate different parts of the sub-manifolds due to the one-prong vs two-prong difference.

Given these strong correlations, we can either train the network to extract the relevant information after extracting the correlations, or we can provide the network with a decorrelated input constructed from the first eight EFPs. To stabilize the training and to save training time we choose the second option and use principle component analysis (PCA). For this we first subtract the mean of the distribution from each data point. We then change to the eigenvalue base of the covariance matrix, removing all linear correlations between the individual components. Finally, we scale the individual components so that their standard deviation is one. The resulting distribution is as close to a normal distribution as is possible using a linear transformation.

## 3 K-means

Anomaly searches are not limited to deep learning. One can also employ a variety of classical ML methods, in particular for density estimation. However, density estimation for multidimensional data, like jet images, is notoriously difficult. Standard approaches like kernel methods or density estimation based on histograms scale badly with the data dimensions $d$ and the number of training points $n$.

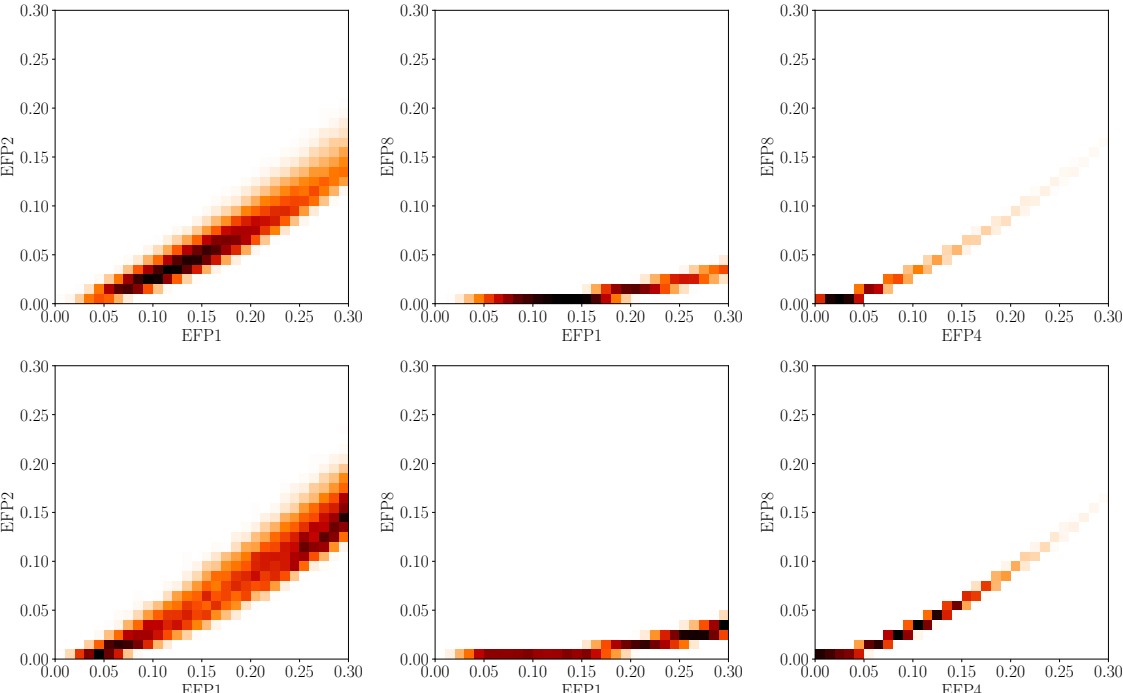

Figure 5: Sample correlations among some of the EFPs for QCD jets (upper) and the Heidelberg dark jets (lower).

For anomaly detection, it is not crucial to know the actual density. Any anomaly score which is (strongly) correlated with the density can potentially be useful. We use the well known k-means clustering algorithm to define such anomaly scores. Loyd's k-means algorithm [89] scales linearly with the number of data points and dimensions for each iteration. Since it usually converges quickly [90] (in our application $\sim 300$ iterations are sufficient for convergence), we can apply it to large datasets with high-dimensional data. K-means provides a given number $k$ of clusters with centroids

$$\vec{\mu}_i = \frac{1}{N_i} \sum_j \vec{r}_{i,j}, \tag{10}$$

where the vectors $\vec{r}_{i,j}$ represent the data instances $j$ assigned to cluster $i$, and $N_i$ is the number of data instances in cluster $i$. The clusters divide the data into a patchwork of Voronoi cells. We use the Lloyd's k-means algorithm with 10 different initializations of the centroids following the "k-means++" prescription [91] and pick the one with the lowest inertia, as implemented in the scikit-learn python library [92].

K-means is neither a density estimation nor an anomaly detection algorithm. However, assigning an effective size to each cluster $i$ around its centroid, e.g.

$$\rho_i = \frac{1}{N_i} \sum_j |\vec{r}_{i,j} - \vec{\mu}_i| \tag{11}$$

(with $j$ iterating over the vectors assigned to cluster $i$), the clusters map out the large-scale data distribution, and we can construct several useful anomaly scores. In this context, using many clusters seems to be beneficial to approximate the underlying distribution as precisely and detailed as possible. On the other hand, the number of data points in each cluster has to be large enough to assign a statistically meaningful size. We employ k-means with $k = 100$ clusters. Using 100000 training images, the smallest cluster contains 78 jet images, and all

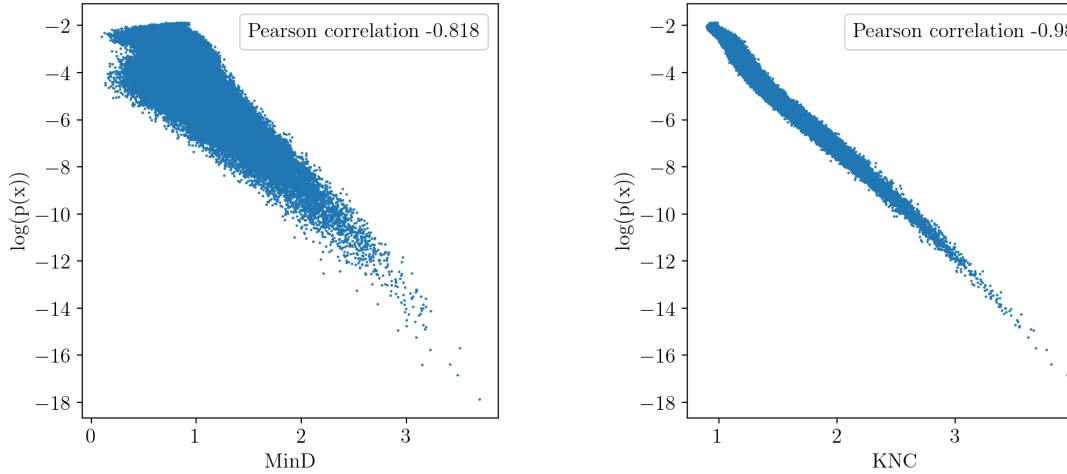

Figure 6: The logarithmic density as a function of MinD (left) and KNC5 (right) for a data set drawn from a five-dimensional normal distribution with unit covariance. All 100000 training points are shown.

others more than 100 jet images. All our models are trained using $40 \times 40$ jet images that are convoluted with a Gaussian smearing kernel with $\sigma = 3$ pixels.

In addition to the datasets described in Sec. 2, we also test the anomaly-detection methods discussed in this section on the standard benchmark set for top tagging [93]. It consists of QCD and top jets with $p_T = 550 \ldots 650$ GeV, in contrast to the low $p_T = 150 \ldots 300$ GeV QCD jets used for our dark jets. The jet constituents are processed into images in the same way as described in Sec. 2. Following the discussion in Refs. [3,4], we take QCD jets as a background and top jets as the anomalous signal (direct top tagging) as well as top jets as a background and QCD jets as the signal (reverse top tagging). These additional tagging examples help to illustrate the differences between different anomaly scores based on k-means.

## 3.1 K-nearest centroids

A simple anomaly score for a jet image, which does not take into account the cluster sizes, is the minimal distance to one of the k-means cluster centroids. We refer to this anomaly score as MinD. A similar approach was discussed in Ref. [55], where instead of k-means clustering a k-medoids algorithm was used to obtain the representatives of the background dataset. The MinD score assumes that regular datapoints are close to the k-means centroids, whereas outliers are not. However, MinD has several obvious drawbacks. K-means itself is susceptible to outliers far from the main distribution, since it may assign a cluster to a single outlier or a small group of outliers. Obviously, the number of clusters is a crucial parameter in this context. Moreover, points on the boundary between two clusters have a higher anomaly score than points close to a cluster center, even if both clusters are part of smooth distributions. For our dataset, we expect such a smooth distribution rather than a collection of well separated clusters.

Both problems can be mitigated by using a score based on $k$ nearest neighbors. In the standard method the distance of a point to its $k$ nearest neighbors is used to estimate the probability density at this point or to determine its affiliation with a class of points. Here, we do not consider the distance to the $k$ nearest data instances but to the $k$ nearest cluster centers (KNC). We define the anomaly score KNC5 as the average distance of a data point to the $k = 5$ nearest centroids obtained through k-means clustering.

Fig. 6 shows MinD and KNC5 for 100 k-means clusters on a data set with 100000 training points drawn from a five-dimensional normal distribution with unit covariance. We observe a strong correlation between MinD and the logarithm of the density. As expected, KNC5 improves the correlation significantly. Although this simple example might not be representative for more complex data distributions, we expect that KNC5 provides a better correlation of the anomaly score and the density than MinD for most smooth distributions.

## 3.2 Gaussian mixture model

As a benchmark for our k-means method we use a Gaussian mixture model (GMM), a standard density estimation technique where the density is approximated by a sum of several multidimensional Gaussians. The GMM uses an iterative expectation maximization algorithm to fit the Gaussians to the data distribution. On the one hand, the GMM provides another simple benchmark. On the other hand, a GMM is an obvious generalization of k-means based algorithms since a GMM provides a mean for each of the mixture components. These means are equivalent to the k-means centroids if one uses an expectation maximization with a covariance matrix proportional to the unit matrix and a common variance, which approaches zero [94]. In this limit, each data point is assigned to one of the mixture components as it is assigned to one cluster in k-means.

We use the scikit-learn python library [92] to fit a GMM to our jet images smeared with our standard Gaussian kernel with $\sigma = 3$. Usually all entries of the covariance matrix in a GMM are fit parameters. However, the 1600 dimensions of the jet images, with a $1600 \times 1600$ covariance matrix, are prohibitive for the full fit. Instead, we use spherically symmetric Gaussians with a variance $\alpha_i + \beta$ for each mixture $i$, where $\alpha_i$ is a fit parameter and $\beta$ is a regularization parameter. We decrease $\beta$ until the smallest fitted variance reaches $10^3 \times \beta$. This way, we ensure that $\beta$ does not dominate the variance, as it would be the case if we used the default scikit-learn parameter $\beta = 10^{-6}$.

For smooth distributions one would expect the $\sqrt{\alpha_i}$ to be of the same order of magnitude as the typical length scale of the dataset, e.g. the $\rho_i$ of the k-means clusters (see Tab. 1). However, the GMM finds $\sqrt{\alpha_i}$ which are roughly two orders of magnitude smaller. This mismatch is due to the fact that the effective dimension of our data is much smaller than 1600, implying the data lives in a lower-dimensional subspace. Fitting a spherical Gaussian to the data that has almost zero variance in many of its dimensions will result in a strongly underestimated standard deviation. Hence, the actual likelihood estimation is very poor. Nevertheless, using the negative log-likelihood of the GMM to define the GMMLL anomaly score might still be valuable. The insight regarding the effective dimension of the data distribution motivates a new density-based anomaly score using k-means clustering, introduced next.

## 3.3 Likelihood-inspired anomaly scores

Building a regularly shaped histogram to estimate the density of our 1600-dimensional data space is of course impossible due to the curse of dimensionality. Instead, we propose to use the k-means clusters as generalized bins which are automatically adapted to the underlying distribution, in analogy to the Gaussians centered around their means in the GMM.

To be specific, we approximate each cluster as a multidimensional sphere with radius $\rho_i$ around its centroid $\vec{\mu}_i$. In the spirit of density estimation, we associate a likelihood to find a

data instance $\vec{r}$ associated to cluster $i$ according to

$$
L_i(\vec{r}) = \mathcal{N}_i
\begin{cases}
1, & \text{for} \quad |\vec{r} - \vec{\mu}_i| < \rho_i, \\[2mm]
\left( \dfrac{\rho_i}{|\vec{r}|} \right)^{d-1} \exp\left[ -\dfrac{(|\vec{r} - \vec{\mu}_i| - \rho_i)^2}{2\sigma_i^2} \right], & \text{for} \quad |\vec{r} - \vec{\mu}_i| > \rho_i,
\end{cases}
$$

$$
\text{with} \quad \sigma_i^2 = \frac{1}{N_i} \sum_j (|\vec{r}_{i,j} - \vec{\mu}_i| - \rho_i)^2, \tag{12}
$$

with $j$ iterating over vectors assigned to cluster $i$. Here, $\mathcal{N}_i$ is a normalization factor and $d$ an effective dimension to be discussed below. Inside the cluster we assume a constant density. However, the clusters are not taken as spheres with sharp boundaries, but we add Gaussian tails such that outliers have different scores depending on the distance to the cluster border. The tails also ensure that the likelihoods of points in the gap between two close, but not overlapping clusters can add up. The factor in front of the Gaussian is chosen such that the marginal one-dimensional likelihood $L_i(|\vec{r}_{i,j} - \vec{\mu}_i|)$ resembles the observed distribution of the data instances. The factor is proportional to the inverse of the volume factor connecting $L_i(\vec{r})$ and $L_i(|\vec{r}_{i,j} - \vec{\mu}_i|)$ in a space with dimension $d$.

For density estimation based on histograms, the normalization is defined as $\mathcal{N}_i = N_i/V_i$, where $V_i$ denotes the volume of the bin. In our multidimensional problem it is non-trivial to estimate the volume $V_i$. One option is $V_i \propto \rho_i^{1600}$. As a consequence, two clusters with only slightly different radii $\rho_i \neq \rho_j$ would have extremely different densities inside the clusters. However, in Sec. 3.2, we have already discussed that the effective dimensions of the subspace where the data lives is much smaller.

For a sensible normalization, one has to approximately determine the effective number of dimensions. Making the simplifying assumption that the underlying density $n$ of data points in and around a cluster is uniform, a sphere of radius $R$ in $d$ dimensions contains $N(R) = n\pi^{d/2}R^d/\Gamma(d/2 + 1)$ data points. This equation can be solved for the dimensionality of the dataset $d_i(R) = \ln(N_i(R)/N_i(cR))/\ln(c)$ at a scale $R$, where $N_i(R)$ is the number of training points in the sphere of radius $R$ around the cluster centroid, and $c$ is a scaling factor

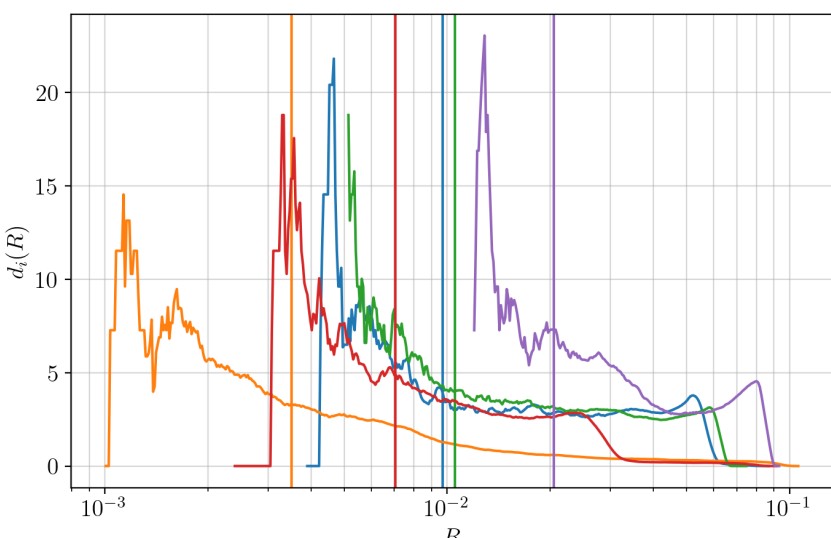

Figure 7: Effective dimensionality $d(R)$ for five different high-$p_T$ QCD clusters. The vertical lines are located at $R = \rho_i$ for each cluster. The intersection point is the estimate for the effective dimension of the cluster.

which we set to $c = 1.1$. The relevant length scale for each cluster is assumed to be $\rho_i$, such that the effective dimensionality of a cluster is defined as $d_i(\rho_i)$. Fig. 7 shows $d_i(R)$ for five clusters of the high-$p_T$ QCD images of the top-tagging dataset together with the corresponding $\rho_i$ values. The effective dimension is estimated in a region with sufficient statistics, where the curve is rather smooth. This means we have chosen $c$ sufficiently large for sufficient statistics, but also small enough to access the shape of $d_i(R)$. We notice that as $R$ approaches the size of the actual distribution as a whole, its dimensionality decreases to zero, since any distribution looks point-like from a large distance. As shown in Tab. 1, the median values for the effective dimensionality of the k-means clusters range between 5 and 8, depending on the data preprocessing. The same value $d = \text{med}\,(d_i(\rho_i))$ is then used to calculate $L_i(\vec{r})$ for all clusters.

Our anomaly tagging algorithm can be summarized as:

1. Perform k-means clustering on a dataset, using $k = 100$.

2. Compute $\rho_i$, $\sigma_i$, and $d_i = d(\rho_i)$ for each cluster $i$ from the distribution of points inside cluster $i$.

3. Find $d = \text{med}\,(d_i)$ as a representative effective dimension for all clusters.

4. Compute the normalization factor $\mathcal{N}_i$ by requiring $\int_{\mathcal{R}^d} L_i(\vec{r})d^d\vec{r} = N_i$, i.e. the likelihood is integrated in $d$ dimensions.

5. For each data point $\vec{r}$ compute $L_i(\vec{r})$ for each cluster as defined in Eq.(12).

6. Compute the anomaly score $-\log(L(\vec{r}))$ with $L(\vec{r}) = \sum_{i=0}^{k} L_i(\vec{r})$.

The corresponding anomaly score is called MLLED (k-Means based Log-Likelihood estimation in Effective Dimensions).

For the QCD jets in our background datasets, we find cluster radii $r_i$ which vary by up to one order of magnitude, i.e. we find a variation of likelihoods by a factor of roughly $10^5$ using the estimated effective dimensions. A point in one of the smallest clusters has a density roughly $10^5$ larger than a point in a cluster which is 10 times larger. Hence, a data point has to be a few sigmas away from the small cluster to have the same small likelihood as a data point in the large cluster. The higher the dimensionality of the clusters the more we put weight on assigning high anomaly scores or low likelihoods to the points in the low-density clusters, as compared to the points that are outliers of the small and highly populated clusters. If the dimensionality is too high, the dependence of the score on the cluster size will dominate over the dependence on the distance to the cluster border. Hence, out-of-cluster anomalies that might lie only a few sigmas away from the border cannot be distinguished anymore from

Table 1: Properties of the set of 100 clusters found by k-means for the different datasets used as background for anomaly tagging.

| | $\min \rho_i$ | $\text{med}\,\rho_i$ | $\max \rho_i$ | $\max \dfrac{\sigma_i}{\rho_i}$ | $\text{med}\,d_i(\rho_i)$ |
|---|---|---|---|---|---|
| high $p_T$ QCD | 0.0031 | 0.011 | 0.026 | 0.32 | 5.2 |
| high $p_T$ top | 0.0127 | 0.015 | 0.026 | 0.27 | 5.4 |
| low $p_T$ QCD | 0.0040 | 0.012 | 0.022 | 0.28 | 5.8 |
| low $p_T$ QCD with $\sqrt[4]{p_T}$ | 0.0088 | 0.012 | 0.018 | 0.25 | 7.8 |

probably non-anomalous points inside smaller clusters. Accordingly, the tagging performance for such anomalies is strongly diminished. As we will see, this is the case for tagging dark jets from the Aachen dataset or QCD jets in a top-jet background. On the other hand, some types of anomalies reside in the low but non-zero background density region of the data space. In these cases the correct hierarchy of clusters and thus a significant dependence on the cluster size and population is required. Such examples may include the tagging of top jets in a QCD background (reverse top tagging).

According to the previous discussion, different normalization factors in Eq.(12) may lead to anomaly scores being more or less sensitive to different types of anomalies. Hence, we also consider an anomaly score based on the unrealistic assumption $d = 1$. This is no longer an estimate for the density, but might still perform well as an anomaly score to tag out-of-cluster outliers with good precision while preserving a certain hierarchy for the in-cluster densities. We refer to this anomaly score as MLL1D.

For k-means the radius of a cluster $\rho_i$ is anti-correlated with the number of data points $N_i$ in the cluster. Taking only $\rho_i$ or $N_i$ into account for the normalization may thus suffice for a qualitatively correct ordering of their densities. Using $d = 1$ but $\mathcal{N}_i = N_i$ instead of the choice defined above defines the MLLN anomaly score.

## 3.4 Performance

After defining the different k-means anomaly scores, we discuss their performance on our benchmark datasets. Top jets have a prominent 3-prong structure and do not require special preprocessing in order to highlight the features relevant for top-tagging in a QCD background. On the other hand, QCD jets and dark jets are dominated by the intensity in the central pixels. In these cases it is promising to apply preprocessing to highlight the dim features, for instance using a $p_T^{1/4}$ reweighting. It is evident from Tab. 1 that such a preprocessing also increases the number of effective dimensions of the clusters, as the number of relevant pixels increases.

The ROC curves for direct and reverse top tagging are shown in Fig. 8, and for tagging dark-jet anomalies in Fig. 9. We see that MinD, KNC5, MLLED and MGG all have similar performances, except for the reverse top tagging case. Ignoring the cluster size completely in assigning a density, MLLN has a low performance on the top-signal dataset, but it performs

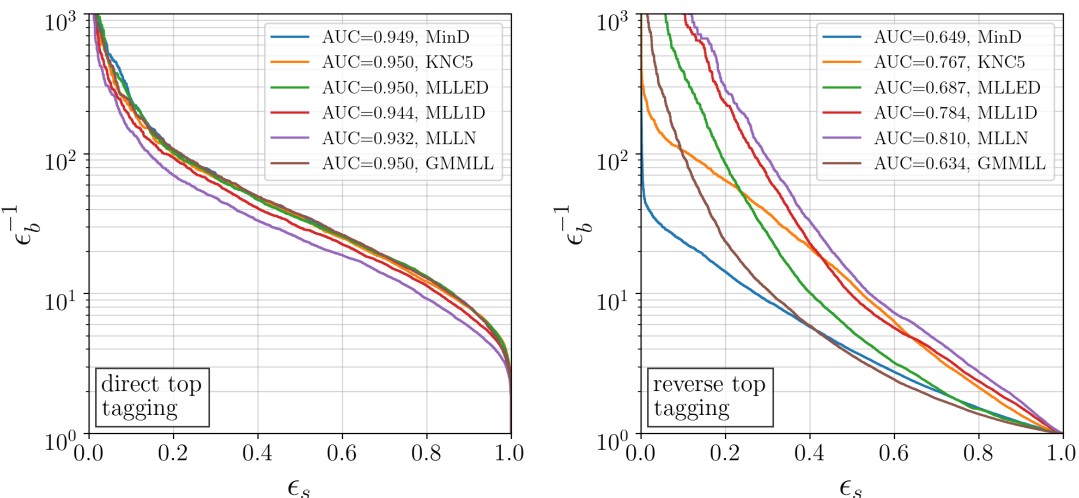

Figure 8: ROC curves for tagging high-$p_T$ top jets in a QCD background (left) and high-$p_T$ QCD jets in a top background (right). The various anomaly scores are discussed in the text.

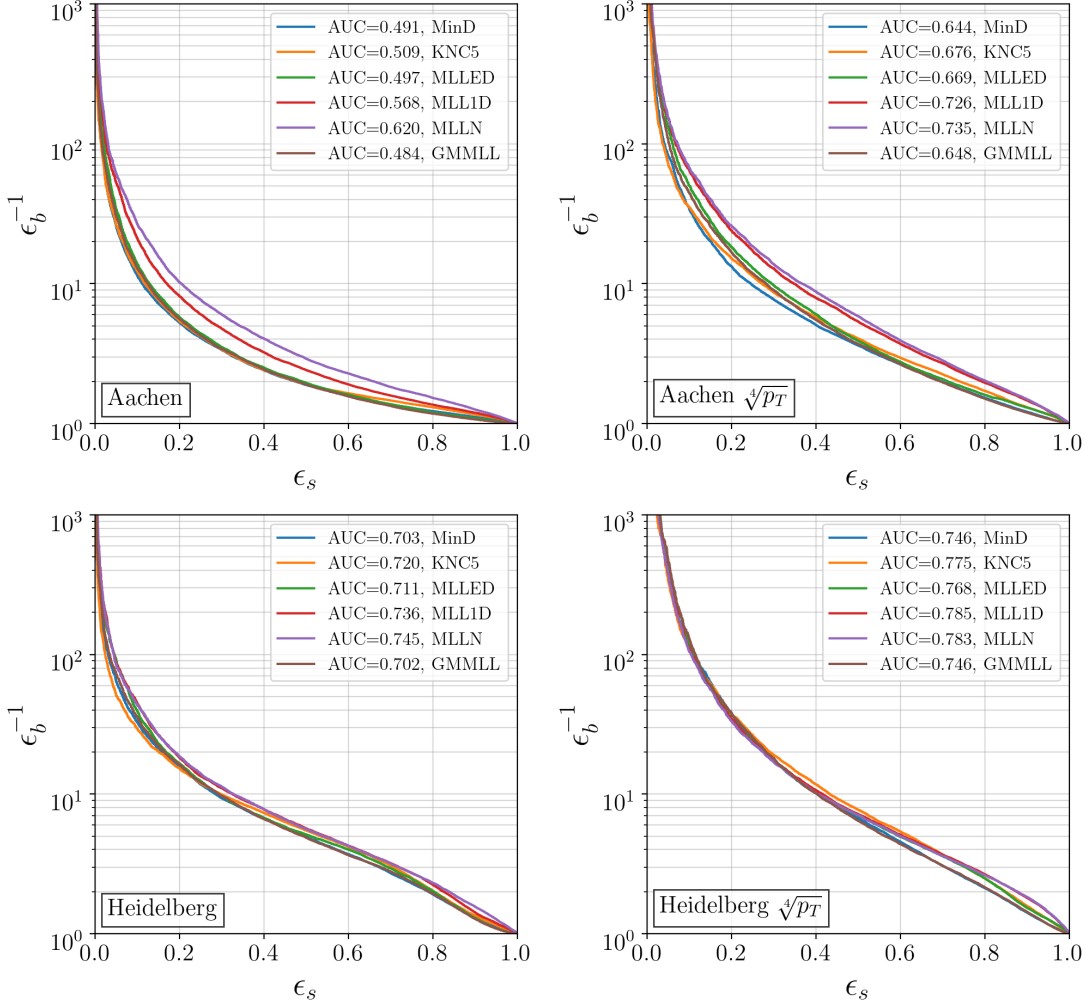

Figure 9: ROC curves for tagging Aachen (top row) and Heidelberg (bottom row) dark jets in a QCD background. The pixel intensities in the images are given by the $p_T$ within a pixel (left) or $p_T^{1/4}$ (right). The various anomaly scores are discussed in the text.

best on the Aachen dataset and for reverse top tagging. The reason for this, as discussed at the end of Sec.. 3.3, is that MLLED and MLLN are sensitive to different kinds of anomalies. MLL1D is a compromise between MLLED and MLLN. It does not show the best, but a reliable performance on all five tasks. This renders MLL1D the most model-agnostic anomaly detection algorithm in this section.

In line with what we find for the anomaly detection methods presented in the following chapters, we observe a strong improvement in the dark jet tagging performance after pre-processing the jet images. The Gaussian filter combined with the 4th root reweighting are essential for the applicability of the MSE distance measure on which our clustering and density estimation algorithms rely. Using $p_T$ instead of $p_T^{1/4}$ implies neglecting the contribution of the low-$p_T$ pixels in the distance measure between images, whereas performing no filtering will result in a distance being artificially large for images with a small spatial shift of intensity.

# 4 Dirichlet-VAE

If the latent space of a variational autoencoder (VAE) is expected to encode physical information on the structure of a jet, then the choice of latent space is important. A comparison of Gaussian, Gaussian Mixture, and Dirichlet latent spaces for detecting anomalous top jets in a QCD sample and vice versa shows that the Dirichlet VAE (DVAE) achieves the best performance and provides an intuitive physical interpretation of how the jets are organised in latent space [4]. The Dirichlet distribution is a family of continuous multivariate probability distributions with the probability density

$$\mathcal{D}_\alpha(r) = \frac{\Gamma\left(\sum_i \alpha_i\right)}{\prod_i \Gamma(\alpha_i)} \prod_i r_i^{\alpha_i - 1} \qquad (i = 1 \ldots R), \tag{13}$$

where $R$ is the number of latent space dimensions. It is a compact and potentially multimodal distribution, constrained such that $\sum_i r_i = 1$, and is defined by $R$ hyper-parameters $\alpha_i > 0$. The hyper-parameters allow us to build hierarchies into the latent space, defined by $\langle r_i \rangle = \alpha_i / \sum_i \alpha_i$. With $\alpha_i < 1$ the distribution has $R$ modes with peaks at each of the latent-space points $r_i = 1$. This enables the DVAE to separate the jets into different, potentially hierarchical, modes based on their kinematics. The DVAE can be thought of as a more powerful deep-learning edition of more traditional topic models, such as LDA, which have also been applied to anomaly detection problems in high-energy physics [12, 29].

In a forward-pass through the network we need to be able to sample from the Dirichlet distribution and calculate the KL-divergence between two Dirichlet distributions. This is very difficult with the exact form of the distribution. Therefore, we use a softmax approximation to the Dirichlet distribution [8]

$$r_i \sim \text{softmax}\,\mathcal{N}(z; \tilde{\mu}, \tilde{\sigma}), \qquad \text{with} \qquad \tilde{\mu}_i = \log \alpha_i - \frac{1}{R}\sum_j \log \alpha_j,$$

$$\text{and} \qquad \tilde{\sigma}_i = \frac{1}{\alpha_i}\left(1 - \frac{2}{R}\right) + \frac{1}{R^2}\sum_j \frac{1}{\alpha_j}. \tag{14}$$

The DVAE loss function is the sum of the reconstruction loss and the KL-divergence between the prior and the latent distribution for each jet,

$$\mathcal{L} = -\langle \log p_\theta(x|r) \rangle_{q_\phi(r|x)} + \beta_{\text{KL}} D_{\text{KL}}\left(q_\phi(r|x), \mathcal{D}_\alpha(r)\right), \tag{15}$$

with a learnable encoder $q_\phi(r|x)$ and decoder $p_\theta(x|r)$, where $\phi$ and $\theta$ are the respective parameters. The reconstruction loss is computed as the KL-divergence between the input and reconstructed jet images, and the KL-divergence between the prior and the latent-representation of the jet becomes

$$D_{\text{KL}}\left(q_\phi(r|x), \mathcal{D}_\alpha(r)\right) = \frac{1}{2}\sum_{i=1}^{R}\left(\frac{\sigma_i^2}{\tilde{\sigma}_i^2} + \frac{(\tilde{\mu}_i - \mu_i)^2}{\tilde{\sigma}_i^2} - 1 - \log \frac{\sigma_i^2}{\tilde{\sigma}_i^2}\right). \tag{16}$$

Here, $\mu_i$ and $\sigma_i$ are the encoded means and variances in the softmax-Dirichlet approximation for each jet.

The DVAE architecture used here is identical to the one in Ref. [4]. The encoder is a neural network with 1600 inputs, a flattened $40 \times 40$ image, with $2R$ outputs with linear activations, and a single hidden layer of 100 nodes with SeLU [95] activations. These outputs are the means and variances used to sample from the softmax-Dirichlet distribution and to calculate the KL-divergence with the prior. The $R$-dimensional vector sampled from the softmax-Dirichlet distribution is then passed to a decoder network which has a very simple architecture; a 1600-dimensional output with no hidden layers and no biases. A softmax activation is applied to the output layer.

## 4.1 Anomaly scores

The Dirichlet latent space allows the VAE to separate the jets in latent space, based on their phase-space features. Because of the mixture model interpretation of the DVAE and simple decoder architecture, we can also visualise the features that the network associates with each mixture. From Ref. [4] we know that when the DVAE is trained on equal parts top and QCD jets, the mixtures are associated with one-prong QCD-like and three-prong top-like jets. In contrast, for datasets with predominantly QCD jets the learned features are one-prong and two-prong jets.

We can choose between three anomaly scores for the DVAE [4]; the reconstruction error, the KL-divergence, or a latent coordinate $r_i$. The coordinate $r_i$ is only unambiguous for $R = 2$, since $r_1 = 1 - r_0$, while for $R > 2$ there is more than one option for the direction. In this work we use $R = 2$ with $\alpha = (1.0, 0.1)$, unless otherwise specified. We train the DVAE solely on background jets and use the reconstruction loss as the anomaly metric, which we expect to be correlated with the density of the jets in physics space. We use the Adam optimizer [96] with a learning rate of 0.01 and decay rates $\beta_{1,2} = (0.9, 0.99)$. The model is trained for 300 epochs, which is sufficient for the loss to converge. We also choose $\beta_{KL} = 0.1$, so that the prior has a large impact on the training. The exact details of the implementation and training are laid out in Tab. 2, and match those described in Ref. [4].

## 4.2 Jet image performance

We find that the preprocessing of the jet images has a large effect on the anomaly detection performance, with different preprocessing parameters being optimal for different signals. For the Aachen dataset we find that $p_T$-reweighting, $p_T \rightarrow p_T^n$, with $n \lesssim 0.1$ and a Gaussian filter with width $\sigma \simeq 1.0$ work best, resulting in an AUC $\simeq 0.71$ and a $\epsilon_b^{-1}(\epsilon_s = 0.2) \simeq 37$. While for the Heidelberg dataset we find that $n \simeq 0.4 - 0.6$ and $\sigma \simeq 1.0$ tends to work best, with an AUC of $\simeq 0.73$ and a background suppression of $\epsilon_b^{-1}(\epsilon_s = 0.2) \simeq 27$. It is certainly not ideal that different anomalies are best detected with different preprocessings, although this is a recurring theme of this paper and we argue in Sec. 2 that this can be understood. We also note that these results do not require a fine-tuning of the preprocessing parameters, as they are largely insensitive to order-one changes.

The ROC curves for the DVAE are summarized in Fig. 10. We see that the Gaussian filter preprocessing has a large effect, especially for the Aachen dataset, where the AUC drops below 0.5 in some cases without the Gaussian filter. This is because the anomalous features in the Aachen dataset are very sparse and at low $p_T$, so the DVAE can and will ignore these unless they are explicitly emphasized in the input. We also studied a higher-dimensional latent space,

Table 2: Network and training parameters for the DVAE.

| Parameter | Value |
|---|---|
| training data set size | 100k |
| number of epochs | 300 |
| batch size | 2048 |
| initial learning rate | $10^{-2}$ |
| $\beta_{KL}$ | 0.1 |
| $\alpha$ | (1.0,0.1) |
| optimizer | Adam |
| $\beta_{1,2}$ | (0.9,0.99) |

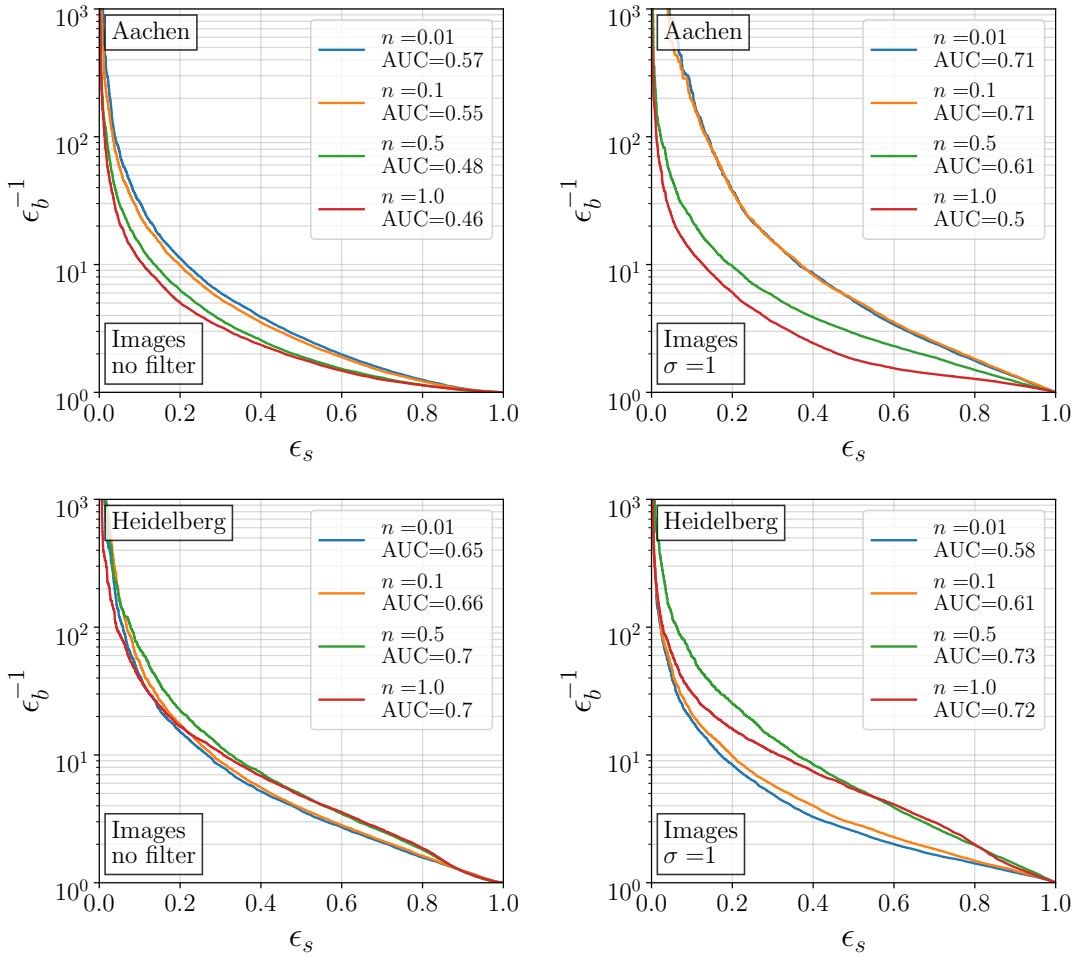

Figure 10: ROC curves for DVAE anomaly detection using jet images on the Aachen and Heidelberg datasets. The reweighting power $n$ is defined in Eq.(9).

$R = 3$, with $\alpha = (1.0, 0.25, 0.1)$, and found no difference in the performance. In [4] it was found that for the top tagging the mixture weights or latent coordinates $r_i$ can provide good performance in anomaly detection even when the anomalous jets are less complex than the background. Here we find a similar behaviour, but the performance is not as good as for the reconstruction error.

## 4.3 EFP performance

The DVAE aims to extract physical information from the images, but it is of course possible to use EFPs as input rather than images. This way it might be easier for the algorithm to characterize the background and better identify anomalies.

As discussed in Sec. 2.3 we consider two preprocessings for EFPs: (i) standard scaling where we set the mean to zero and the standard deviation to unity, not removing the strong correlation between EFPs, and (ii) using PCA components, still with zero mean, but an identity covariance matrix between features. These are performed with the StandardScaler and PCA routines in scikit-learn [92]. We have also learned from the previous section that reweighting physical inputs heavily influences the anomaly detection performance. Here we investigate what happens when we train our DVAE on the first eight EFPs and we preprocess them using $z_i \rightarrow z_i^{\kappa}$ and $R_{ij} \rightarrow R_{ij}^{\beta}$, see Eq.(7), in analogy to the pixel reweighting used before. We explore

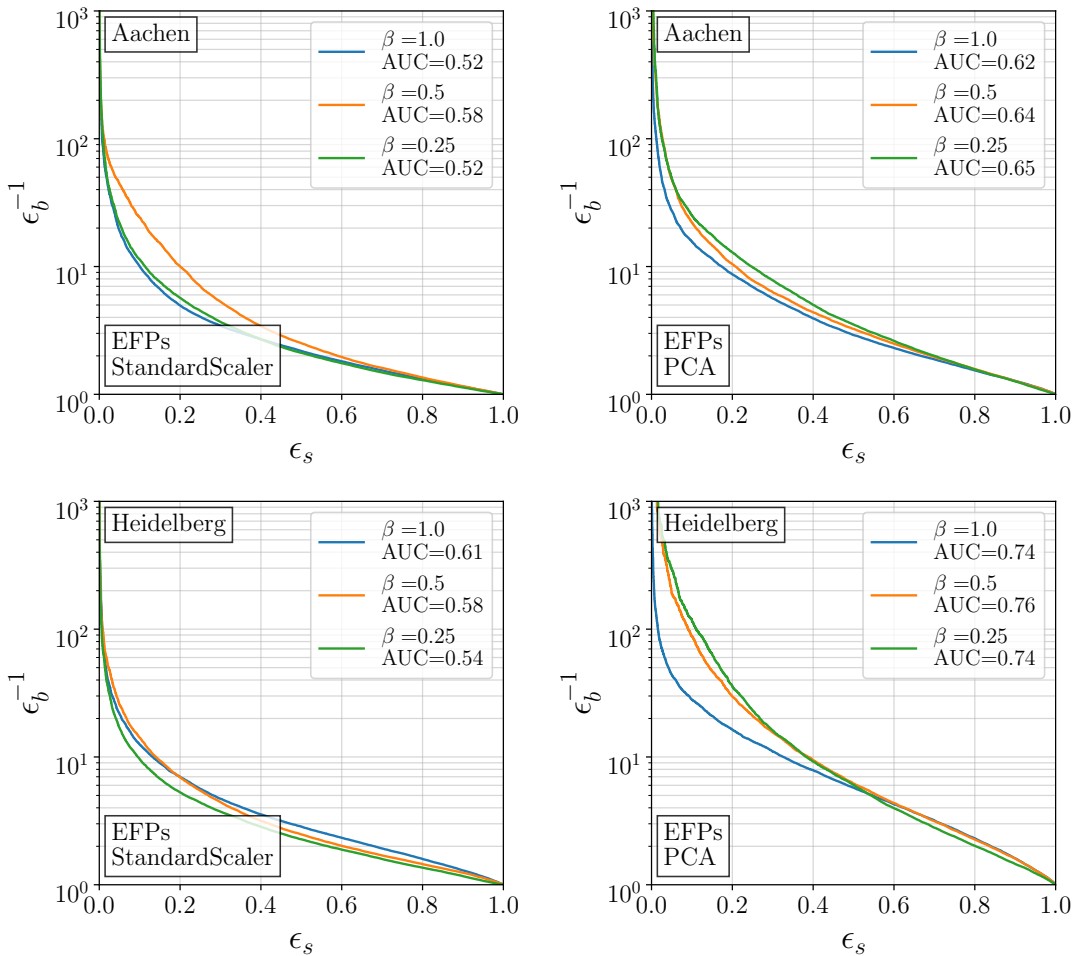

Figure 11: ROC curves for DVAE anomaly detection using EFPs on the Aachen and Heidelberg datasets ($\kappa = 1$).

a range of values for $\beta$ and $\kappa$, as well as having either two or three mixtures in the latent space, i.e. $R = 2$ or $R = 3$. In agreement with the DVAE trained on images, we find that the results with $R = 2$ and $R = 3$ are essentially the same. A summary of the DVAE performance for $\kappa = 1$ and $\beta = 0.25 \ldots 1$ are shown in Fig. 11. First, we see that standard scaling without decorrelating the EFPs essentially fails. Once we include the decorrelation step, we find that for both the Aachen and Heidelberg datasets using EFPs with $\kappa = 1$ and $\beta = 0.25 \ldots 0.5$ gives the best anomaly detection. This is in complete agreement with our observation from k-means clustering and the image-based DVAE. Directly comparing the performance of the image-based and EFP-based DVAEs we find that in our current setup the images work slightly better for the Aachen dataset, while the EFPs are more efficient in extracting the hard substructure of the Heidelberg dataset.

# 5 INN

A density estimation based on neural networks can be obtained using normalizing flows [68, 69], specifically their invertible neural network (INN) incarnation [70, 71]. INNs are neural networks which learn bijective mappings between a physics and a latent space completely symmetrically in both directions. They allow access to the Jacobian and both directions of the

mapping, linking density estimation in the physics and latent spaces in a completely controlled manner. We have used the flexible INN setup successfully for precision event generation [97, 98], unfolding detector effects [99], and QCD or astro-particle inference [100, 101].

For this straightforward application we use simple, affine coupling blocks [68] combined with random, but fixed, orthogonal transformations. In an affine coupling block, the input dimensions are split into two halves, $x_{1,2}$. The first half is passed through a subnet which learns two functions $s(x_1)$ and $t(x_1)$. The second half is transformed by an element-wise multiplication ($\odot$) and an element-wise addition,

$$\begin{pmatrix} z_1 \\ z_2 \end{pmatrix} = \begin{pmatrix} x_1 \\ x_2 \odot e^{s(x_1)} + t(x_1) \end{pmatrix} \quad \Leftrightarrow \quad \begin{pmatrix} x_1 \\ x_2 \end{pmatrix} = \begin{pmatrix} z_1 \\ (z_2 - t(z_1)) \odot e^{-s(z_1)} \end{pmatrix}. \tag{17}$$

The Jacobian of this mapping is

$$J = \begin{pmatrix} \mathbb{1} & 0 \\ \partial_{x_1} z_2 & \text{diag } e^{s(x_1)} \end{pmatrix} \quad \Rightarrow \quad \log |J| = \sum s(x_1). \tag{18}$$

We use soft clamping to avoid instabilities in the training [70], replacing $s(x_1) \to 2 \tanh s(x_1)$ in Eqs.(17) and (18). The random transformations ensure that in each coupling block the information is split differently. They are easily invertible, and their Jacobian is one.

If $f$ is the mapping between physics space and the INN latent space, and $q$ is the prior distribution in latent space, then the learned distribution in physics space $p$ can be written as $p(x) = q(f(x))\,|J(x)|$. The loss function should become minimal if the learned distribution in physics space $p$ matches the true distribution $p_{\text{true}}$. Therefore, we would like to minimize the KL-divergence between $p$ and $p_{\text{true}}$,

$$D_{\text{KL}}(p, p_{\text{true}}) = \int dx\, p_{\text{true}}(x) \, \log \frac{p_{\text{true}}(x)}{p(x)} \,. \tag{19}$$

Since we know $p_{\text{true}}$ only from samples $\{x_i\}$, this is difficult to achieve. Moreover, we can split the KL-divergence into the self-entropy of $p_{\text{true}}$, which does not depend on $f$, and the negative log-likelihood. The loss function is this negative log-likelihood,

$$\mathcal{L} = -\int dx\, p_{\text{true}}(x) \, \log p(x) = -\int dx\, p_{\text{true}}(x) \Big[ \log q(f(x)) + \log |J(x)| \Big]$$

$$\approx -\frac{1}{N} \sum_{i=1}^{N} \Big[ \log q(f(x_i)) + \log |J(x_i)| \Big], \tag{20}$$

Table 3: Network and training parameters for the INN.

| Parameter | Value |
|---|---|
| training data set size | 100k |
| number of epochs | 300 |
| batch size | 512 |
| initial learning rate | $10^{-4}$ |
| learning rate decay | 0.98 |
| initial noise width | 0.5 |
| noise decay | 0.95 |
| optimizer | Adam |
| $\beta_{1,2}$ | $(0.9, 0.999)$ |
| scaling | PCA |

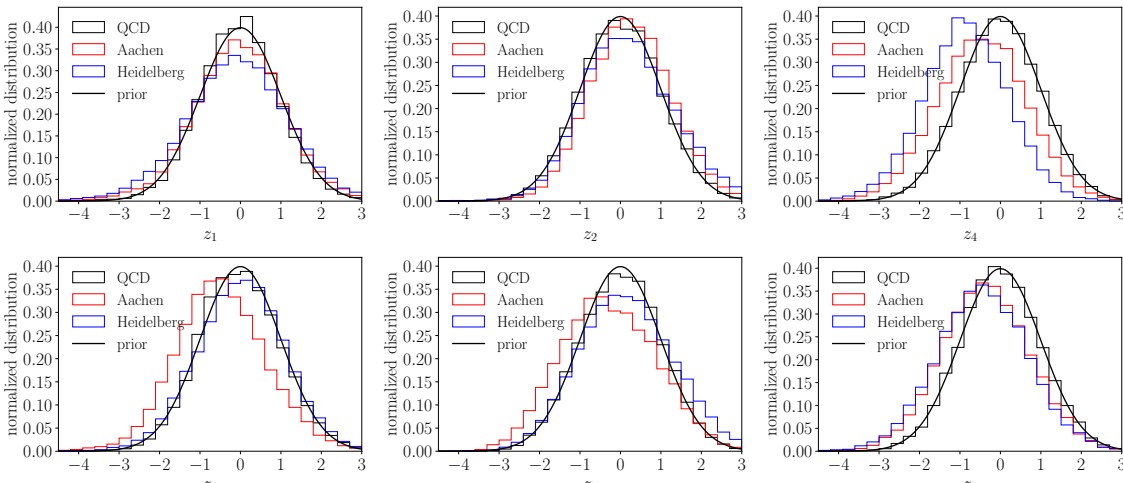

Figure 12: Example distribution of background and signal jets in the INN latent space, after training on background only.

which we can evaluate without having an explicit form of $p_{\text{true}}$. This loss function automatically ensures that the latent distribution follows the prior [70].

Learning a density from jet images faces the challenge that the active pixels are distributed very sparsely, which means that most of the large number of physics space dimensions do not carry any information. This inflates the number of dimensions in the image-pixel space, while we know that the relevant number of dimensions describing the jets is much smaller. For the bijective INN we are interested in limiting the number of physical and latent dimensions, so we simplify our task by using the eight EFPs up to order $d = 3$ introduced in Sec. 2. The 8-dimensional physics space is then mapped on an 8-dimensional Gaussian latent space. Our architecture consists of 24 affine coupling blocks, each followed by a random orthogonal transformation. Our subnets are each a fully connected network with one hidden layer of 512 nodes and ReLu activation. The output layer has no activation. The network parameters are summarized in Tab. 3.

We train our network for 300 epochs using the Adam optimizer [96] and an initial learning rate of $10^{-4}$. The learning rate is then reduced every epoch by 2%. To stabilize the training, we apply a PCA as described in Sec. 2.3. Since the PCA is also an invertible transformation with a computable Jacobian, we can still evaluate the density in the original EFP space. Also we help the training by adding Gaussian noise to the training data, where we reduce the standard deviation of the noise by 5% every epoch, so that it had no effect by the end of the training. Further training details can be found in Tab. 3.

## 5.1 Anomaly scores

The INN allows us to estimate the density associated to where a particular jet lies in the space of physical observables we use to characterize it. It does this by constructing a Gaussian latent space for the jets along with a learned Jacobian to correctly account for the density in the transformation. The idea behind using an INN to search for anomalous jets is that we can look for jets located in low density regions of either physics space or the Gaussian latent space, with the Jacobian connecting the two.

Using the density of a given jet in physics space as the anomaly score, based on information from both the latent Gaussian space and the Jacobian is well motivated and has a clear definition. It can also be interpreted as a standard observable, defined in physics space, but numerically represented through a neural network benefiting from a specific latent space.

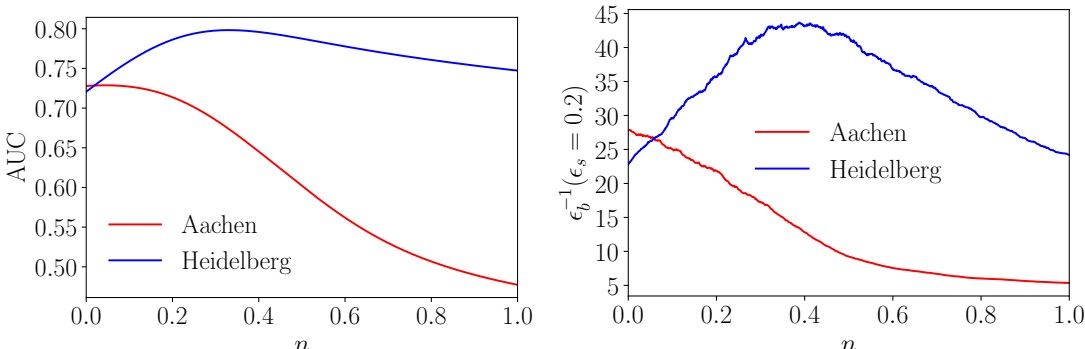

Figure 13: Effect of the input scaling $x \to x^n$ on the AUC and the inverse mistag rate at 20% signal efficiency.

In the latent space, we expect the INN to ensure that the background forms a Gaussian distribution, including the exponentially suppressed tails away from the mean, or typical jet patterns. Thus we can use the distance to the center as anomaly score, which is monotonously related to the likelihood for a multi-dimensional Gaussian. In Fig. 12 we show the latent space distribution for the QCD training dataset and the Aachen and Heidelberg dark jets. While the QCD distributions follow the Gaussian prior closely, as expected when training on QCD jets only, both signal datasets differ from the prior in some of the latent directions. The drawback of this method is that the training does not guarantee the signal to end up in the tales of the latent distribution.

## 5.2 Performance

Applying the INN to EFPs with $\kappa = \beta = 1.0$ and using the negative log-likelihood in physics space as an anomaly score leads to a strong bias to detect more complex jets as anomalous. This is due to the fact that jet density is particularly high at low EFP values and jets with less structure have lower EFP values. This bias can be compensated for by using a reweighting that leads to more uniform distributions. We study an element-wise exponentiation $g(x) = x^n$ applied to the EFP before the PCA, so the distribution in the $x'$-space is $p'(g(x)) = p(x) \cdot |J_g(x)|^{-1}$. The Jacobian of this reweighting is again diagonal, and the negative log-likelihood of a single sample transforms like

$$\mathcal{L} \to \mathcal{L} + \sum \log \left| \frac{dg}{dx} \right| = \mathcal{L} + (n-1) \sum \log x + \sum \log n \,, \tag{21}$$

where the sums go over the eight input dimensions. We drop the last term since it is independent of $x$ and will therefore have no effect on tagging. Then for $n = 0$ this transformation corresponds to the reweighting $g(x) = \log(x)$. This technique for reweighting would be analogous to training the DVAE, or any autoencoder, on reweighted inputs, and then applying the reweighting to the input and reconstructed images when evaluating the anomaly score.

Fig. 13 shows the effect of this reweighting for $n \in [0, 1]$ on the AUC and the inverse mistag rate at 20% signal efficiency. The point $n = 1$ means no reweighting at all. It can be seen that a signal with less structure than the background like the Aachen data set needs low values of $n$ to be detected, as expected. Fig. 14 shows the ROC curves for density based tagging in physics and in latent space. Only the density in physics space is affected by the reweighting. For the plots we choose three representative values for $n$. Despite the INN not being retrained on EFPs with different reweightings, the anomaly detection performance drastically improves for different $n$. This implies that the INN is learning the low-density regions in physics space

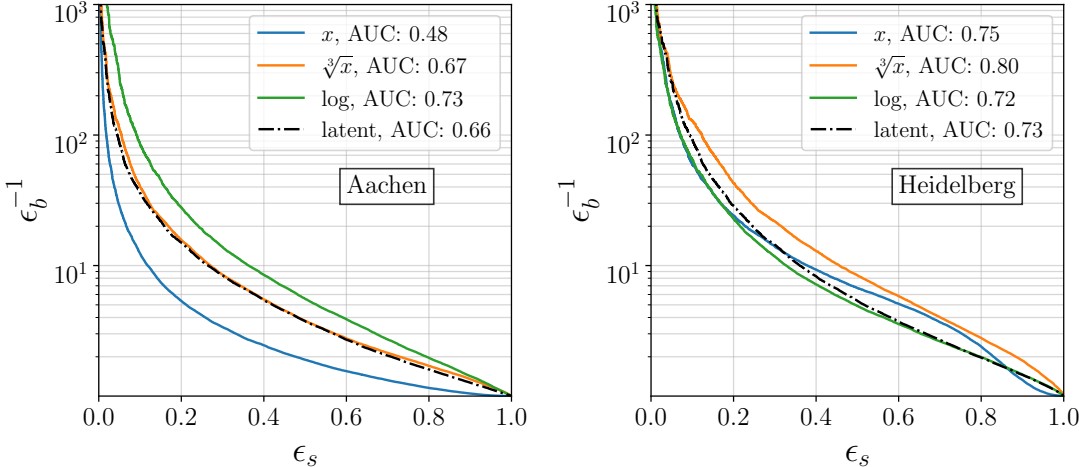

Figure 14: ROC curves for INN anomaly detection on the Aachen and Heidelberg datasets, using density based tagging in physics and latent space. The density in physics space is effected by reweighting while the density in latent space is not.

well, and that difficulties in detecting the anomalies are related to the choice of observables rather than the network training. A remaining challenge for the INN-based anomaly search is the choice of the reweighting, as the log-reweighting works best for the semi-visible Aachen dataset, while the cubic-root-reweighting gives the best results for the Heidelberg dataset.

# 6 Outlook

Searches for anomalies, or hints for physics beyond the Standard Model, are a large part of the LHC program and essentially all particle physics experiments. We have studied how this analysis goal can be pursued with modern machine learning tools and concepts, specifically unsupervised learning. New ways to disentangle potential signals from background can be applied at the LHC in many ways, at the trigger and analysis stages, on jets or other specific analysis objects and on whole events, with training on data and on simulation.

In this paper we have studied jets, which are an excellent path to understanding modern anomaly search tools, because they are theoretically well understood, can be simulated in large numbers, and come with huge established data samples. We do know that classic out-of-distribution searches will not work in LHC physics, because in high-statistics jet or event samples the backgrounds eventually populates all possible phase space configurations. This suggests to consider anomaly searches in relation to probability density estimation. This task is reflected in our choice of two benchmark signals, both inspired by a dark or invisible sector interfering with QCD showering. The Aachen dataset features semi-visible jets, where part of the shower products are not visible, while the Heidelberg dataset features massive decays inside a fully visible jet.

We studied three different ways of searching for such dark jets in a QCD background sample, assuming that we can train in an unsupervised way on background only or with negligible signal contamination:

· K-means is a classic ML algorithm, which we use to define a set of different anomaly scores. It works on jet images and allows us to study density estimation in the corresponding high-dimensional spaces. For a well-motivated choice of hyperparameters the performance is stable and competitive with more modern deep learning methods.

Anomaly searches with k-means clustering benefit from a reweighting of the image pixels as part of the preprocessing, as do all methods we have analysed.

· Dirichlet VAEs are a modern ML-algorithm which constructs a multi-modal latent space, allowing us to define anomaly scores in the physics space or in the latent space. The reconstruction loss is used as the anomaly score, which is expected to be correlated with the density of the jets in physics space. They work on jet images and on lower-dimensional representations like energy flow polynomials (EFPs). We found that Dirichlet VAEs also require preprocessing, and while it is possible to use a latent-space anomaly score, the reconstruction loss performs better for our benchmark signals.

· Normalizing flows, specifically INNs, construct a bijective and traceable map between the physics and latent spaces. This means they are ideally suited for density estimation but they work best on lower-dimensional data representations, like decorrelated EFPs. INNs have the advantage that we have full control over the link between the two spaces, and we found the density estimate in physics space to provide the best-performing anomaly score.

Our three very different algorithms have very different requirements, strengths, and caveats. Once understood, they all show similar performance on both datasets. Preprocessing is important for all of them, and has a very significant impact on the performances. This is because we define the anomalies as those jets that lie in low density regions of physics space, and the preprocessing alters this density, therefore it changes how the anomalies are defined. We finally note that our Aachen and Heidelberg datasets are challenging datasets for anomaly detection, compared for example to top-tagging, which has served as a benchmark for assessing the performance ML methods. The datasets will be available as public benchmarks for existing and improved anomaly detection algorithms.

# Acknowledgments

**Funding information**    This research is supported by the Deutsche Forschungsgemeinschaft (DFG, German Research Foundation) under grant 396021762 – TRR 257: *Particle Physics Phenomenology after the Higgs Discovery*, grant 400140256 - GRK 2497: *The physics of the heaviest particles at the Large Hardon Collider* and through Germany's Excellence Strategy EXC 2181/1 - 390900948 (the Heidelberg STRUCTURES Excellence Cluster).

# A   Notes

The codes for the k-means clustering, the DVAE and the INN can be found at https://github.com/IvanOleksiyuk/jet-k-means, https://github.com/bmdillon/jet-mixture-vae and https://github.com/ThorstenBuss/jet-inn, respectively.

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
