# Peer review of "What's Anomalous in LHC Jets?"

_SciPost Physics, doi:SciPost Phys. 15, 168 (2023)_

## Round 1 · Referee Report · Anonymous (Referee 1) · 2022-9-12

Report

Let me apologize to the authors for the delay in submitting this referee report. I had erroneously thought I had already submitted my report at the end of August, but clearly did not.

In any case, the authors have responded to all of my concerns in an adequate way. I am happy to recommend publication in one of the SciPost journals.

Whether it should be SciPost Physics or SciPost Physics Core, I leave as a decision to the editor. I have no concerns about the scientific content of the work, and the authors indeed show state of the art results for certain tasks. My concern about the utility of implicit hypothesis testing remains, though the authors understand this limitation of their studied methods.

---

## Round 1 · Referee Report · Anonymous (Referee 2) · 2022-10-23

Report

Thank you for taking into account my feedback and I am very sorry for the delayed response! I have now read the full paper and have a number of comments that should be addressed before I can recommend publication in SciPost Physics. Please note that I still think the intro/conclusions could be improved, in particular in relation to what specifically is studied in the paper.

Major:

  • This paper includes a number of interesting comparisons, but the connection between the framing and the results could be improved. At its core, this paper studies a number of unsupervised methods (none of which are new I believe) for particular BSM signals and I think it would be helpful to frame the paper in that way, instead of over claiming some deeper/broader goal. I would state early on somewhere that the goal of this paper is to examine classical and deep machine learning methods and how preprocessing affects their sensitivity for a class of intersting/difficult BSM models.
  • Sec. 2.3: Don't both your standardization and de-sparcification actually reduce the information content of the jets? What is the tradeoff between loss in sensitivity from information distortion and ease of training for the NN? (maybe the former could be quantified using fully supervised networks?)

Minor:

  • "Looking at LHC physics these concepts can be developed most easily for QCD jets" -> I agree these are available in large numbers and are less complex than full events, but they are more complex than everything else. Maybe this should say that jets are useful because they are so complex that it is "easy" for BSM to hide in them?
  • Sec. 1, paragraph 2: this paragraph comes out of nowhere! It would be useful to give a bit more context before diving into details of AEs. For example, "supervision" is not defined. I would also avoid "is the simplest" which is a matter of opinion and I think most people would even agree it is not true (I would say things like kNN is even simpler). Paragraph 4 also seems to imply ("Based on these practical successes") that the AE work came first in the story of anomaly detection in HEP, which is not true.
  • Sec. 1, paragraph 3: "is mapped to a low-dimensional latent distribution" -> this is not quite correct; for VAEs, the latent space need not be smaller than the input space (although in practice, it usually is).
  • "the concept of outliers is not defined," -> this seems too strong. Just because p > 0 for everything doesn't mean you can't have outliers. Please consider rephrasing.
  • "a simple, working definition of anomalous data is an event which lies in a low-density phase space region" -> this is a fine (albeit coordinate-dependent) definition, but it is not the only one. You discuss other methods later in this paragraph, but you seem to imply that they are variations of the low density assumption (which is not true). Please consider rephrasing. This is also an issue of the next paragraph, e.g. "This way, anomaly searches are fundamentally linked to density estimation" is also not true for all AD searches.
  • Intro: Pythia is mentioned without a citation.
  • "Dirichlet VAEs (DVAEs) outperform for example standard VAEs" -> I don't think you know this generically - please clarify that this has been shown for specific models (this is a problem with model independent searches!)
  • Intro: why call them INNs? Don't you just mean normalizing flows? (also no citation here). It is fine to use a particular NF implementation of course, but it seems strange to specialize already here.
  • "The bottom line of our comparison is that density estimation really is at the basis of ML- based anomaly searches in LHC jets." -> please rephrase (see above).
  • validity: -
  • significance: -
  • originality: -
  • clarity: -
  • formatting: -
  • grammar: -

Author:  Thorben Finke  on 2023-08-14  [id 3904]

(in reply to Report 2 on 2022-10-23)

We thank the referee for their additional comments and feedback and apologise for the delay in responding. We address the various comments below:

Major:

  • This paper includes a number of interesting comparisons, but the connection between the framing and the results could be improved. At its core, this paper studies a number of unsupervised methods (none of which are new I believe) for particular BSM signals and I think it would be helpful to frame the paper in that way, instead of over claiming some deeper/broader goal. I would state early on somewhere that the goal of this paper is to examine classical and deep machine learning methods and how preprocessing affects their sensitivity for a class of intersting/difficult BSM models.

We added "specific examples" to the abstract and changed the first section title to "Introduction". Together with the other changes as per the minor revisions below, we believe that the overall tone is now no longer over claiming a broader goal.

  • Sec. 2.3: Don't both your standardization and de-sparcification actually reduce the information content of the jets? What is the tradeoff between loss in sensitivity from information distortion and ease of training for the NN? (maybe the former could be quantified using fully supervised networks?)

Since the standardization is performed image wise, it does indeed destroy information. However, the preprocessing steps are chosen to remove mainly redundant information and have been shown to improve classifier performance, since not all jet orientations need to be learned explicitly. Dividing by the total pT removes the different pT distributions as a trivial discriminator and improves the numerical performance, as stated in section 2.3. The further rescaling by pT^n; n in (0, 1] is bijective and thus does not destroy any further information. The de-sparcification uses a fixed Gaussian kernel, i.e. there is no random smearing. Thus no information is lost, except for edge effects where intensity is smeared beyond the image. We expect this loss to be negligible, as the phi and eta ranges of the images are sufficiently large compared to the jet radius.

We added a clarification in section 2.3.

Minor:

  • "Looking at LHC physics these concepts can be developed most easily for QCD jets" -> I agree these are available in large numbers and are less complex than full events, but they are more complex than everything else. Maybe this should say that jets are useful because they are so complex that it is "easy" for BSM to hide in them?

We added a sentence to that phrase: "Nevertheless, they are complex enough such that possible new physics signatures can hide in a non-trivial way."

  • Sec. 1, paragraph 2: this paragraph comes out of nowhere! It would be useful to give a bit more context before diving into details of AEs. For example, "supervision" is not defined. I would also avoid "is the simplest" which is a matter of opinion and I think most people would even agree it is not true (I would say things like kNN is even simpler). Paragraph 4 also seems to imply ("Based on these practical successes") that the AE work came first in the story of anomaly detection in HEP, which is not true.

We added a sentence for a smoother transition to this paragraph, stating the difference between supervised and unsupervised methods, and why the unsupervised methods are favored. We softened the statement about AEs and removed the statement "Based on these practical successes". We want to note that this was not meant to ignore other works on anomaly detection in HEP, but rather specific to deep learning approaches for unsupervised anomaly detection.

  • Sec. 1, paragraph 3: "is mapped to a low-dimensional latent distribution" -> this is not quite correct; for VAEs, the latent space need not be smaller than the input space (although in practice, it usually is).

We added "typically" to this phrase. We think that a further discussion on the distinction between over- and undercomplete AEs would not benefit the content and would harm the flow of reading.

  • "the concept of outliers is not defined," -> this seems too strong. Just because p > 0 for everything doesn't mean you can't have outliers. Please consider rephrasing.

We rephrased it: "the concept of outliers is difficult to define unambiguously", as setting e.g. a threshold in p always inserts some ambiguity to the definition of the term outlier.

  • "a simple, working definition of anomalous data is an event which lies in a low-density phase space region" -> this is a fine (albeit coordinate-dependent) definition, but it is not the only one. You discuss other methods later in this paragraph, but you seem to imply that they are variations of the low density assumption (which is not true). Please consider rephrasing. This is also an issue of the next paragraph, e.g. "This way, anomaly searches are fundamentally linked to density estimation" is also not true for all AD searches.

In the first statement, together with the context of difficulty to define outliers in the first place, we do not claim this to be the only definition of outliers as it is "a" simple definition. We agree that the other methods indeed have another definition of what an anomaly is. Therefore, we rephrased the transition to the CWoLa method as an alternative definition of anomalies as overdensities. As for the next paragraph, we rephrased it such that it is less general and points towards the methods applied in this work.

  • Intro: Pythia is mentioned without a citation.

Citation added in the introduction.

  • "Dirichlet VAEs (DVAEs) outperform for example standard VAEs" -> I don't think you know this generically - please clarify that this has been shown for specific models (this is a problem with model independent searches!)

We rephrased accordingly that the DVAE outperforms the other methods on the task of finding hadronically decaying, boosted top jets.

  • Intro: why call them INNs? Don't you just mean normalizing flows? (also no citation here). It is fine to use a particular NF implementation of course, but it seems strange to specialize already here.

We call them INN already at this place, to be consistent with later sections. The overall reference as INN is to be consistent with previous literature, e.g. arXiv 2006.06685. We added the citations from the INN section also to the introduction.

  • "The bottom line of our comparison is that density estimation really is at the basis of ML- based anomaly searches in LHC jets." -> please rephrase (see above).

We removed this sentence, as it is specific to our methods used and this specific relation to density estimation is discussed afterwards.

Having addressed all the concerns of the referee and having included the corresponding changes in the draft, we think that the paper is ready for publication.

---

## Editorial Decision

published